Cranial anatomy of Bolotridon frerensis, an enigmatic cynodont from the Middle Triassic of South Africa, and its phylogenetic significance

Pusch Luisa C. 1 2 luisa.pusch@mfn.berlin
http://orcid.org/0000-0002-0596-623X Kammerer Christian F. 3 4
http://orcid.org/0000-0002-2501-9387 Fröbisch Jörg 1 2 4
1 Museum für Naturkunde, Leibniz-Institut für Evolutions- und Biodiversitätsforschung , Berlin , Germany
2 Institut für Biologie, Humboldt-Universität zu Berlin , Berlin , Germany
3 North Carolina Museum of Natural Sciences , Raleigh, NC , USA
4 Evolutionary Studies Institute, University of the Witwatersrand , Johannesburg , South Africa
Hedrick Brandon
Electronic publication date: 2021 Jun 16
Publication date: 2021
Volume: 9
Electronic Location ID: e11542
Received 2021 Mar 24; Accepted 2021 May 10
Copyright: © 2021 Pusch et al.
Copyright year: 2021
Copyright holder: Pusch et al.
License: This is an open access article distributed under the terms of the Creative Commons Attribution License, which permits unrestricted use, distribution, reproduction and adaptation in any medium and for any purpose provided that it is properly attributed. For attribution, the original author(s), title, publication source (PeerJ) and either DOI or URL of the article must be cited.
License URL: https://creativecommons.org/licenses/by/4.0/

Keywords: Therapsida, Computed tomography, Endocranial anatomy, Phylogeny

Funding: Deutsche Forschungsgemeinschaft: (FR 2457/8-1) This work was supported by a grant from the Deutsche Forschungsgemeinschaft (FR 2457/8-1) to Jörg Fröbisch. The funders had no role in study design, data collection and analysis, decision to publish, or preparation of the manuscript.

==============================
The cynodont fauna of the Trirachodon-Kannemeyeria Subzone of the Middle Triassic Cynognathus Assemblage Zone (AZ) is almost exclusively represented by taxa belonging to the clade Eucynodontia. However, there is one basal (non-eucynodont) cynodont known to have survived into this assemblage: the enigmatic Bolotridon frerensis. BSPG 1934-VIII-7 represents by far the most extensive specimen of B. frerensis, consisting of a partial skull with occluded lower jaw. The specimen was initially described by Broili & Schröder (1934), but their description was limited to surface details of the skull and the dental morphology. Here, by using a computed tomographic (CT) reconstruction, we redescribe this specimen, providing novel information on its palatal and internal anatomy. New endocranial characters recognized for this taxon include ridges in the nasal cavity indicating the presence of cartilaginous respiratory turbinals. New data obtained from the CT scan were incorporated into the most recently published data matrix of early non-mammalian cynodonts to test the previously unstable phylogenetic position of Bolotridon. Our phylogenetic analyses recovered Bolotridon as the sister-taxon of Eucynodontia, a more crownward position than previously hypothesized.

Introduction

Cynodontia is one of the six major therapsid subclades and was the last to diversify, first appearing in the fossil record in the late Permian but not showing substantial species richness until the Middle Triassic, when it became the most diverse and abundant synapsid group (Hopson & Kitching, 1972; Kemp, 2005; Abdala & Ribeiro, 2010; Angielczyk & Kammerer, 2018; Lukic-Walther et al., 2019). In the form of their mammalian descendants, cynodonts remained the most successful synapsid lineage throughout the Mesozoic and onwards to the present day. By contrast, early in their history cynodonts are a minor component of terrestrial ecosystems, and with a few notable exceptions (e.g., Galesaurus, Thrinaxodon) are rare and poorly known from late Permian and earliest Triassic faunas. However, recent work has improved our knowledge of early cynodont diversity. Abdala et al. (2019) described a new, surprisingly large cynodont taxon (Vetusodon elikhulu) from the late Permian (Lopingian) of the Karoo Basin in South Africa, and Huttenlocker & Sidor (2020) described another new taxon (Nshimbodon muchingaensis) from the upper Permian Madumabisa Mudstone Formation of Zambia. In addition to Vetusodon, five further Permian cynodont taxa are currently recognized from the Karoo Basin: Charassognathus gracilis and Abdalodon diastematicus (the oldest known cynodonts, currently known from a single skull each from the upper Endothiodon Assemblage Zone (AZ)), Procynosuchus (a broadly-distributed taxon with additional records from elsewhere in Africa and in Europe), and Cynosaurus suppostus and Nanictosaurus kitchingi. The latter two species may represent early representatives of primarily Triassic lineages, suggesting that the cynodont radiation had begun prior to the end-Permian mass extinction (Broom, 1937; Kemp, 1979; Van Heerden & Rubidge, 1990; Abdala & Allinson, 2005; Botha, Abdala & Smith, 2007; Kammerer, 2016; Van den Brandt & Abdala, 2018; Abdala et al., 2019; Viglietti, 2020).

None of the late Permian genera are known to have survived the end-Permian mass extinction, but at least two lineages (Galesauridae and Thrinaxodontidae) seem to have crossed the Permo-Triassic boundary, and were fairly abundant in the Lower Triassic Lystrosaurus declivis AZ of the Karoo Basin in South Africa (Botha & Smith, 2020). Galesaurids are represented in the Triassic by the taxa Galesaurus and Progalesaurus (e.g., Sidor & Smith, 2004; Pusch, Kammerer & Fröbisch, 2019), whereas Thrinaxodontidae is represented by the extremely common and extensively studied taxon Thrinaxodon (e.g., Fourie, 1974; Rowe, Carlson & Bottorff, 1995; Abdala, Jasinoski & Fernandez, 2013; Jasinoski, Abdala & Fernandez, 2015). A rarer representative of early cynodonts in the Lystrosaurus declivis AZ is the unusually broad-skulled genus Platycraniellus (Abdala, 2007). By the Middle Triassic, these taxa had disappeared, with later Triassic cynodonts consisting almost exclusively of members of the diverse clade Eucynodontia. At least seven genera of eucynodonts are recorded in the Middle Triassic Cynognathus AZ (and additional, undescribed taxa are also probably present; see Hendrickx et al. (2020): the cynognathians Cynognathus, Diademodon, Langbergia, Cricodon, and Trirachodon, the probainognathian Lumkuia, and the phylogenetically uncertain taxon Cistecynodon (Brink & Kitching, 1953; Hopson & Kitching, 2001; Abdala & Giannini, 2002; Sidor & Smith, 2004; Abdala, Hancox & Neveling, 2005; Abdala & Ribeiro, 2010). Most of these genera (all but Langbergia and Cricodon) are known from the most extensively-exposed and best-studied portion of the Cynognathus AZ, the Trirachodon-Kannemeyeria Subzone (Hancox, Neveling & Rubidge, 2020).

One additional cynodont taxon is currently recognized from the Trirachodon-Kannemeyeria Subzone, the enigmatic Bolotridon frerensis. Preliminary studies have suggested that Bolotridon (=Tribolodon; see Coad (1977)) is the only basal (i.e., non-eucynodont) cynodont to survive into the Middle Triassic (Sidor & Smith, 2004; Abdala, Hancox & Neveling, 2005; Abdala & Ribeiro, 2010). However, the rarity and incompleteness of the known material of this taxon have made it difficult to resolve its phylogenetic position, which has been tested computationally in only one analysis (Sidor & Smith, 2004). Hopson & Kitching (1972) regarded Bolotridon as a member of the family Galesauridae, albeit in a pre-cladistic sense containing all non-eucynodont epicynodonts, rather than an indication of any particular relationship to Galesaurus. Battail (1991) included Bolotridon in the family Thrinaxodontidae based on the possible presence of a complete osseous palate. However, the presence of a complete secondary palate has never been confirmed for Bolotridon. The earliest definite presence of this mammal-like feature in cynodonts occurs in thrinaxodontids, and this character is then retained in all crownward taxa (Fourie, 1974; Van Heerden & Rubidge, 1990; Hopson, 1991; Crompton, Musinsky & Owerkowicz, 2015; Crompton et al., 2017). An exception to its universal retention has recently been suggested for the Permian Vetusodon, which has an incomplete secondary palate, but was recovered crownward of Thrinaxodon in the initial analysis of Abdala et al. (2019). However, the most recent phylogeny by Huttenlocker & Sidor (2020) supports a more basal position for Vetusodon, placing it with Cynosaurus near the base of Epicynodontia (outside of the origin of the complete secondary palate).

The holotype of Bolotridon frerensis is a partial left dentary (NHMUK PV R2583), which was described by Seeley (1894, 1895) and collected at the historic Cynognathus AZ locality of Lady Frere Commonage (Cacadu, Chris Hani District Municipality, Eastern Cape Province) in South Africa. A right femur and left tibia (NHMUK PV R2584) were found together with the holotype while removing matrix from a skull of “Gomphognathus” (=Diademodon) (Seeley, 1895) and have been referred to B. frerensis as well, but the association of these elements with the holotype jaw is questionable. The majority of other referred specimens from the type locality also only represent isolated dentary or maxillary elements (BSPG 1934-VIII-8, 9, 501, 502, 503, 504, and 505) and do not provide further information on the broader anatomy of this taxon. The sole exception to this is the specimen BSPG 1934-VIII-7, which is by far the most complete known specimen of B. frerensis and consists of a partial skull with occluded lower jaw (Figs. 1–3). Broili & Schröder (1934) provided an initial description of BSPG 1934-VIII-7, but their description was limited to surface details and focused on the dental morphology of this and other referred specimens. The tight occlusion of the upper and lower jaws also prevented the authors from exposing the palatal region without damaging the skull through overpreparation. Accordingly, the morphology of the palate remains largely unknown.

Figure 1 BSPG 1934-VIII-7 in dorsal view.

(A) Photograph and (B) 3D reconstruction of the specimen. Abbreviations: C, upper canine; d, dentary; fr, frontal; ju, jugal; mx, maxilla; na, nasal; pa, parietal; pmx, premaxilla; po, postorbital; pra, prearticular; prf, prefrontal; sq?, squamosal?. Photograph in (A) by Christian F. Kammerer. Image of the 3D reconstructed skull in (B) by Luisa C. Pusch.

Figure 2 BSPG 1934-VIII-7 in right lateral view.

(A) Photograph and (B) 3D reconstruction of the specimen. Abbreviations: C, upper canine; cp, coronoid process; d, dentary; fr, frontal; Is, upper incisors; ju, jugal; la, lacrimal; mf, masseteric fossa; mx, maxilla; na, nasal; pa, parietal; PCs, upper postcanines; pmx, premaxilla; pnp, prenasal process of the premaxilla; po, postorbital; pra, prearticular; prf, prefrontal; sq?, squamosal?. Photograph in (A) by Christian F. Kammerer. Image of the 3D reconstructed skull in (B) by Luisa C. Pusch.

Figure 3 BSPG 1934-VIII-7 in left lateral view.

(A) Photograph and (B) 3D reconstruction of the specimen. Abbreviations: C, upper canine; cp, coronoid process; d, dentary; fr, frontal; Is, upper incisors; ju, jugal; la, lacrimal; mx, maxilla; na, nasal; pa, parietal; PCs, upper postcanines; pmx, premaxilla; pnp, prenasal process of the premaxilla; po, postorbital; pra, prearticular; prf, prefrontal. Photograph in (A) by Christian F. Kammerer. Image of the 3D reconstructed skull in (B) by Luisa C. Pusch.

Here, we present a redescription of Bolotridon frerensis based on computed tomographic (CT) reconstruction of BSPG 1934-VIII-7. This provides new insights into the previously-obscured palatal region and novel information on endocranial characters in this taxon. Although the palate is damaged in BSPG 1934-VIII-7, our redescription provides substantial new data on its anatomy allowing incorporation into the most recent published data matrix of basal cynodonts (Huttenlocker & Sidor, 2020). This new information is used to test the currently unstable phylogenetic placement of Bolotridon (Hopson & Kitching, 1972; Battail, 1991; Sidor & Smith, 2004). We also discuss the faunal composition of the Cynognathus AZ, with a special focus on Lady Frere Commonage, which yields seemingly unusual faunal elements not recorded in other exposures of this assemblage zone.

Materials & Methods

Specimen

The specimen of Bolotridon frerensis described herein (BSPG 1934-VIII-7; Figs. 1–9) is part of the collection of the Bayerische Staatssammlung für Paläontologie und Geologie in Munich. The block containing the fossil remains of this and most other referred specimens of B. frerensis (BSPG 1934-VIII-8, 9, 501, 502, 503, 504, and 505) was collected in the Burgersdorp Formation, in exposures representing the Trirachodon-Kannemeyeria Subzone of the Middle Triassic Cynognathus AZ. It was discovered by G. Grossarth at Lady Frere Commonage in 1931, as part of a research trip by the Bayerische Akademie der Wissenschaften. The specimens were later prepared by the technician G. Kochner of the same institution (Broili & Schröder, 1934). The skull of BSPG 1934-VIII-7 is laterally compressed and cracked, but the dentition and many of the individual cranial elements are in good condition. The total length of the specimen is 8.1 cm and its maximum width is ~3.7 cm.

Figure 4 Mid sagittal section through the reconstructed skull and lower jaw of BSPG 1934-VIII-7, with the vomer and palatine digitally removed to show the internal morphology.

Abbreviations: c, lower canine; cp, coronoid process; d, dentary; fr, frontal; I, upper incisor; i, lower incisor; ju, jugal; la, lacrimal; mx, maxilla; na, nasal; obl, olfactory bulb location; och, olfactory chamber; olac, opening of lacrimal canal; pa, parietal; pcs, lower postcanines; pmx, premaxilla; pnp, prenasal process of the premaxilla; po, postorbital; pra, prearticular; prf, prefrontal; pt, pterygoid; rch, respiratory chamber; rmxt?, ridge for cartilaginous maxilloturbinals?; sp, splenial; sq?, squamosal?, tf, transverse flange of the pterygoid. Image by Luisa C. Pusch

Figure 5 Transverse CT sections through the olfactory chamber (A, B) of BSPG 1934-VIII-7.

Abbreviations: d, dentary; fr, frontal; ju, jugal; la, lacrimal; lac, lacrimal canal; lr, lateral ridge; mr, median ridge; mx, maxilla; na, nasal; np, nasopharyngeal passage; och, olfactory chamber; olac, opening of lacrimal canal; PC, upper postcanine; pc, lower postcanine; pl, palatine; prf, prefrontal; pt, pterygoid; tl, transverse lamina; vo, vomer. CT images in (A) and (B) by Luisa C. Pusch.

Figure 6 3D reconstruction of the skull roof in ventrolateral view.

Asterisk marks the location of the olfactory bulbs of the brain on the ventral surface of the frontals. Abbreviations: fr, frontal; na, nasal; pa, parietal; po, postorbital; prf, prefrontal; rnt, ridge for cartilaginous nasal turbinals. Image by Luisa C. Pusch.

Figure 7 3D reconstruction of the snout of BSPG 1934-VIII-7.

(A) Snout in dorsal aspect with the nasals, lacrimals, prefrontals, postorbitals, frontals, parietal and squamosal digitally removed to expose its internal structure. (B) Ventral view of the palate. Abbreviations: C, upper canine; Is, upper incisors; ju, jugal; mx, maxilla; pa, parietal; PCs, upper postcanines; pl, palatine; pmx, premaxilla; ppmx, palatal process of the maxilla; pt, pterygoid; sc, secondary choana; sq?, squamosal?; vo, vomer. Images in (A) and (B) by Luisa C. Pusch.

Figure 8 3D reconstruction of the lower jaw of BSPG 1934-VIII-7.

(A) Right dorsolateral, (B) dorsal, and (C) ventral view of the mandible. Abbreviations: ang, angular; cp, coronoid process; cs, lower canines; d, dentary; is, lower incisors; pcs, lower postcanines; pra, prearticular; sp, splenial. Images in (A), (B) and (C) by Luisa C. Pusch.

Figure 9 Symphyseal region of BSPG 1934-VIII-7.

(A) Anterior and (B) ventral views of the symphyseal region highlighted with an asterisk. (C) and (D) virtual horizontal CT sections through the symphyseal region. Abbreviations: ang, angular; d, dentary; sy, symphysis. Photograph in (A) by Luisa C. Pusch and in (B) by Christian F. Kammerer. CT images in (C) and (D) by Luisa C. Pusch.

Computed tomography data

BSPG 1934-VIII-7 was CT scanned in the CT-Laboratory of the Museum für Naturkunde Berlin using an YXLON FF35 X-ray CT scanner with a voxel size of 0.0463 mm, a voltage of 115 kV and a current of 220 µA. The visualization of the slices, virtual 3D rendering, and segmentation of selected structures were performed using VGStudio Max 3.3 (Volume Graphics GmbH, Heidelberg, Germany) in the 3D Visualization Laboratory at the Museum für Naturkunde Berlin.

Phylogenetic analyses

BSPG 1934-VIII-7 was coded into the dataset of the most recent phylogenetic analysis of early non-mammalian cynodonts (Huttenlocker & Sidor, 2020) to assess its relationships. The matrix of Huttenlocker & Sidor (2020) was largely modified from those of Abdala (2007) and Kammerer (2016) and contains a few newly adapted characters from other sources. We reduced the number of characters in their matrix from 111 to 110, because one character was present twice (character 27 = 64; we deleted the repeated character 64). The data matrix further comprises 26 therapsid taxa: two gorgonopsians (Aelurognathus and Cyonosaurus) used as outgroups, eight therocephalians, and 16 cynodonts including the newly added Bolotridon. A parsimony analysis was performed in PAUP 4.0a (build 168) (Swofford, 2003), with minimum branch lengths of zero set to collapse. For the analysis, a heuristic search with tree-bisection-reconnection (TBR) branch-swapping was performed, with multistate characters unordered. A bootstrap analysis was conducted based on 1,000 replicates to assess clade support. Additionally, a Bayesian analysis was performed in MrBayes v. 3.2.7 (Ronquist & Huelsenbeck, 2003) using the Markov model (Lewis, 2001) with a gamma distributed rate parameter. The Markov chain Monte Carlo analysis ran for one million generations (run time 15.11 min) until the standard deviation of split frequencies fell below 0.01. The results of both analyses are included into the updated character list (Data S1) and the updated data matrix can be found in Data S2 and S3.

Results

General preservation

The skull of BSPG 1934-VIII-7 is missing the septomaxillae, temporal region, and occiput. It is laterally compressed, with the right half of the skull having been shifted somewhat posteriorly and ventrally and the left half somewhat anteriorly and dorsally (Figs. 1–3). A small fragment of bone is preserved lateral to the coronoid process of the right dentary and posterior to the jugal, which might pertain to the squamosal (Figs. 1, 2, 4) (this element is otherwise not preserved in this specimen). The palate of BSPG 1934-VIII-7 was strongly affected by lateral compression, and resolution of the palate in the CT-scan is not ideal, but the scan still provides substantial information about the morphology of this region that is not visible externally (Figs. 5, 7). The lower jaw of BSPG 1934-VIII-7 consists of the dentaries, right splenial, anterior portion of the left angular, and prearticulars, with the right prearticular being more complete. The left preartcular is preserved only as a small fragment. The left splenial, right angular, and both surangulars, coronoids, and articulars are missing in this specimen (Figs. 1–4, 8).

Snout

The premaxilla is a small bone forming the anteriormost margin of the snout and the anteromedial and ventral margins of the external naris. Its prenasal (intranarial) process, which is slightly damaged ventrally, extends dorsally to contact the nasals (Figs. 1–4). The facial portion of the premaxilla is posteriorly overlain by the maxilla up to the level of the fourth incisor (Figs. 2, 3). Each premaxilla bears four conical, slightly recurved incisors, which appear unusually long in Bolotridon compared to other basal cynodonts (although their apparent length could be an artifact of slipping out of their sockets post mortem). The left incisor row has been slightly shifted forward, with the result that the left I4 is positioned around the same level as the right I3, and the right I1 and left I1, I2 and I3 are very close together. The fourth incisor on both sides is considerably smaller than the preceding three teeth. The right I4 has partially fallen out of its socket and is displaced posteroventrally. Posteromedially, the premaxilla forms a broad plate underlying the anterior portion of the vomer. At the posterolateral margin of this plate, the premaxilla is separated from the maxilla by an oval paracanine fossa accommodating the crown of the lower canine. This fossa is positioned anteromedial to the canine, as is usual for basal cynodonts. Lateral to the paracanine fossa, a short diastema is present between I4 and the upper canine (Figs. 2, 3, 4, 7B).

The dorsal roof of the snout and nasal cavity is formed by elongate nasal bones. The nasals are transversely narrow at their anterior edge, and narrow further (appearing ‘pinched’ in dorsal view) at mid-length where they are overlapped by the dorsal edges of the maxillae before broadening posteriorly. The nasals contact the frontals posteriorly and the lacrimals and prefrontals posterolaterally, around the level of the anterior orbital margin (Figs. 1, 2, 3, 6). In internal view, the ventral margin of the nasal bone projects deeply into the nasal cavity and is slightly overlapped by the anterior and dorsal portion of the lacrimal posteriorly (Figs. 4, 5B). The underside of the nasals is generally concave, but becomes flatter in the region where they contact the lacrimals. It is characterized by a series of ridges extending anteriorly from the interorbital region along the ventral surface of the frontals and continuing onto the nasals. These ridges consist of a median ridge along the midline suture, which weakens anteriorly, and a pair of lateral ridges running parallel to the median ridge (Figs. 5B, 6). Similar ridges are known from a number of therapsids and have been interpreted as attachment points for cartilaginous nasal turbinals, with the median ridge possibly supporting the dorsal edge of a cartilaginous nasal septum (e.g., Kemp, 1979; Hillenius, 1994; Sigurdsen, 2006; Crompton, Musinsky & Owerkowicz, 2015; Crompton et al., 2017; Bendel et al., 2018; Pusch, Kammerer & Fröbisch, 2019; Pusch et al., 2020; Huttenlocker & Sidor, 2020).

The maxilla is a large bone that extends over most of the lateral surface of the snout and forms much of the sidewall of the nasal cavity and the secondary palate (Figs. 1–4, 7). Medially, the two maxillae contact one another on the ventral surface of the snout. Unfortunately, because of strong lateral compression, it is uncertain whether the maxillae naturally contacted one another to form a complete secondary palate. The maxillae are clearly distorted to some degree, as both palatal processes have been shifted against each other, with the right palatal process overlapping the left one (Fig. 7B). A similar case of deformation-induced contact of the medial projections of the maxillae was recently described for the Early Triassic cynodont Galesaurus (Pusch, Kammerer & Fröbisch, 2019) (undistorted Galesaurus skulls indicate that this taxon did not have a complete secondary palate). Despite their damage, based on the proportions of the maxillary projections in Bolotridon, the secondary palate of this taxon must have been at least as extensive as in Galesaurus (i.e., more expansive than in Permian cynodonts such as Procynosuchus, Dvinia, Abdalodon and Vetusodon, in which the medial processes are transversely shorter; Tatarinov, 1968; Kemp, 1979; Hopson & Kitching, 2001; Ivakhnenko, 2013; Kammerer, 2016; Abdala et al., 2019).

Both the external and internal surfaces of the maxillae are generally smooth, although the external surfaces of both maxillae are marked by some slight cracks (Figs. 2, 3, 4). The internal surface of the maxilla is dorsally strongly overlapped by the ventral margin of the nasals and anteriorly it is covered by the palatal process of the premaxilla. Posteriorly, it is overlapped by the anterior margins of the lacrimal and jugal, and also by the dorsolateral portions of the damaged and displaced palatine (Figs. 4, 5, 7A). Three large foramina perforate the external surface of the maxilla above the tooth row, marking the openings of rami from the maxillary canal. Unfortunately, we were unable to resolve the course of the maxillary canal in the CT-scan, as it was damaged due to the strong compression of the snout.

A total of nine right and eleven left upper postcanines are present in BSPG 1934-VIII-7 (Figs. 2, 3, 7B). The number of upper postcanines slightly differs from what was described by Broili & Schröder (1934), who originally counted ten postcanines in the left maxilla. The discrepancy is likely due to the tight occlusion of upper and lower jaws, making it difficult to see the actual number of the teeth without CT data. The postcanines increase in size from the first to the penultimate tooth. The last postcanine is noticeably smaller than the penultimate tooth in the left maxilla. On the right side the last postcanine is damaged, but appears comparable in size to its predecessor, suggesting that the small size of the left last postcanine could be a result of its recent eruption at the time of death. However, several postcanines have damaged crowns, with only the left PC6, PC9, PC10, and PC11 and the right PC5, PC6, and PC8 being completely preserved, showing the tricuspid morphology described by Broili & Schröder (1934). Notable is the morphology of the intact left PC6 and right PC5 and PC6, which appear to naturally have a shorter main cusp than the more posterior teeth, in which the main cusp is more pronounced. This is most clearly visible in the left PC9, PC10, and PC11 (Figs. 2, 3). The presence of two separated lingual cingular cusps cannot be observed for both upper and lower postcanines in BSPG 1934-VIII-7 due to damage, but have been described in other referred specimens of B. frerensis with better preserved teeth (Broili & Schröder, 1934). There is a short diastema present between the upper canine and the anteriormost upper postcanine, with the left canine being relatively well preserved, whereas in the right canine the crown is more damaged. Serrations are absent on all teeth, and distinct carinae are not present on the canines (Figs. 2, 3, 7B, 8).

The anterior margin of the orbit is formed by the lacrimal, which contacts the prefrontal dorsally, the nasal anterodorsally, the maxilla anteriorly and the anterior part of the jugal posteroventrally (Figs. 2, 3). The medial surface of the lacrimal is posteriorly overlapped by the anterior and ventral margins of the prefrontal. Anteriorly, it covers part of the posterior portion of the maxilla. Its medial surface contacts the palatine, although the extent of this contact has been exaggerated due to compression (Fig. 5B). On the floor of the orbit, the posteroventral extension of the lacrimal also contacts the pterygoid and covers the anterior part of the jugal. The lacrimal is perforated posteriorly by two small foramina, through which the nasolacrimal canal originates in the anterior edge of the orbit. The canal runs anteriorly through a ridge on the medial surface of the lacrimal and opens at the level of the sixth lower postcanine above a small maxillary antrum, which was only weakly visible in the scan (Figs. 4, 5). A short ridge appears to extend slightly forward beyond this opening (Fig. 4). In the 3D reconstruction it resembles the structure described for Galesaurus (Pusch, Kammerer & Fröbisch, 2019) and Thrinaxodon (Crompton, 2013). However, we cannot exclude the possibility that this is just a crack caused by damage and compression of the maxilla in that area. Clear evidence of ridges extending forward beyond the opening of the lacrimal canal have so far been provided for only a few non-mammalian cynodonts, with their earliest appearance in Galesaurus and Thrinaxodon, and have been interpreted as forming the base for cartilaginous maxilloturbinals (Fourie, 1974; Hillenius, 1994; Crompton, 2013; Crompton, Musinsky & Owerkowicz, 2015; Crompton et al., 2017; Pusch, Kammerer & Fröbisch, 2019).

The prefrontal contributes to the anterodorsal edge and also partly to the medial wall of the orbit. Its anterodorsal portion contacts the nasal anteriorly and the lacrimal ventrally, and its posterodorsal portion contacts the postorbital posteriorly (Figs. 1, 2, 3, 6). The suture between the prefrontal and nasal is relatively short in BSPG 1934-VIII-7, related to the small size and slender shape of the prefrontal. The contact between the nasal and lacrimal is accordingly somewhat longer than what is usually observed in basal cynodonts, in which the prefrontal tends to be a larger bone occupying more of this region (e.g., Fourie, 1974; Kemp, 1979; Sidor & Smith, 2004; Abdala, 2007; Botha, Abdala & Smith, 2007; Kammerer, 2016; Van den Brandt & Abdala, 2018; Abdala et al., 2019; Pusch, Kammerer & Fröbisch, 2019). The dorsal margin of the prefrontal is curved and projects inwards to overlap the anterolateral process of the lateral flange of the frontal. In internal view, its anterior margin also slightly overlies the lacrimal along its posterior and posteroventral margins (Figs. 1–4, 5A, 6).

The incomplete jugals make up the posterolateral floor of the orbit and the ventral portion of the zygomatic arch. They contact the maxilla anteriorly, the lacrimal anterodorsally and the anterolateral portion of the pterygoid medially. The better preserved left jugal also slightly contacts the posterolateral tip of the palatine. A contact of the jugal with the palatine seems to be naturally present, as has also been described for Galesaurus (Pusch, Kammerer & Fröbisch, 2019). The missing contact between these bones on the right side is likely an artifact of deformation caused by shifting of the skull halves. In internal view, the jugal is anteriorly overlapped by the posteroventral margin of the lacrimal, which is in turn overlapped by the prefrontal. The jugal forms the ventral portion of the postorbital bar, in the form of a dorsally-directed ‘spike’ that would have contacted the postorbital bone (preserved on the left side only). The posterior preserved tip of the right jugal contacts the fragmented piece of bone that might belong to the squamosal (Figs. 1–4, 7).

Skull roof

The frontals are anteroposteriorly elongate bones extending between the nasals anteriorly and the parietals (of which only a fragment is preserved that does not provide much information on the morphology of the bone) posteriorly. Anterolaterally, the frontal’s lateral flange is overlapped by the curved dorsal margin of the prefrontal, and posterolaterally it contacts the dorsal part of the postorbital (Figs. 1–4, 5A). A deep, elongate depression on the ventral surface of the frontals, which is divided by a faint median ridge into left and right compartments, marks the location for the olfactory bulbs of the brain. The lateral and partial anterior limit of the olfactory bulbs is bounded by two ridges, which weaken anteriorly until they meet a pronounced ridge at the midline suture of the frontals marking the hindmost part of the nasal cavity (Figs. 4, 5A, 6). Similar ridges bounding the olfactory bulbs have been described for the basal cynodonts Procynosuchus (Kemp, 1979) and Galesaurus (Pusch, Kammerer & Fröbisch, 2019), as well as the probainognathian Chiniquodon (Kemp, 2009). The posterior limit of the olfactory bulbs is likely indicated by the gradual transition from slight concavity to convexity of the ventral surface of the frontals at the level of the postorbital bar (Figs. 4, 6). The pronounced ridge originating at the midline suture of the frontals just anterior to the depression housing the olfactory bulbs of the brain also marks the posterior limit of the cartilaginous nasal septum, which was likely attached to the median ridge (Crompton, Musinsky & Owerkowicz, 2015; Crompton et al., 2017).

The postorbitals are badly damaged, with only their anterior portion being preserved. The postorbital contributes to the dorsal margin of the orbit by contacting the posterior portion of the prefrontal anterodorsally and the posterior portion of the frontals dorsolaterally. It also would have formed the posterodorsal portion of the postorbital bar, though its descending ramus contacting the jugal is not preserved (Figs. 1, 2, 3, 6).

Palate

The morphology of the strongly compressed and displaced unpaired vomer is difficult to resolve in the scan of BSPG 1934-VIII-7. It is an elongated strip of bone dividing the nasopharyngeal passage at the midline, which appears to be very narrow as a result of lateral compression of the skull halves (Figs. 5, 7). The anterior portion of the vomer, which overlies the palatal process of the premaxilla (Fig. 7A), is usually characterized by having a grooved dorsal surface housing the vomeronasal organ in therapsids (e.g., Hillenius, 2000; Crompton et al., 2017; Kammerer, 2017, Bendel et al., 2018; Pusch, Kammerer & Fröbisch, 2019; Pusch et al., 2020). However, in BSPG 1934-VIII-7 the anterior portion of the vomer is clearly damaged, and the housing for the vomeronasal organ is not evident (Fig. 7A). The posterior portion of the vomer has two lateral ‘wings’, which are not in their natural positions, with the left wing also being bent ventrally due to damage. These wings are wedged posteriorly between the medially directed plates of the palatines, together forming a transverse lamina, which forms the posterior border of the long primary choanae that extend the length of the vomer and open into the nasopharyngeal passage (Figs. 5, 7A), as has also been described for Galesaurus (Pusch, Kammerer & Fröbisch, 2019), Thrinaxodon (Crompton, 2013) and the more crownward eucynodonts Massetognathus and Probainognathus (Crompton, Musinsky & Owerkowicz, 2015; Crompton et al., 2017). As has been suggested by Hillenius (1994), Crompton et al. (2017), and Pusch, Kammerer & Fröbisch (2019), it is likely that the space dorsal to the transverse lamina made up an olfactory chamber, and the large space anterior to the transverse lamina provided a respiratory chamber, which was presumably filled with cartilaginous maxilloturbinals attached to the short ridge running forward beyond the opening of the lacrimal canal (Figs. 4, 5).

The palatines are laterally compressed, resulting in a strong dorsal displacement especially for the left palatine, which almost reaches the nasals (Figs. 5B, 7). Anteriorly, the palatine contacts the maxilla on the secondary palate. Although damaged, the palatine forms much of the floor, lateral wall and roof of the nasopharyngeal passage and part of the posterolateral wall of the olfactory chamber (Figs. 5, 7) (Crompton et al., 2017; Pusch, Kammerer & Fröbisch, 2019). Similar to Galesaurus (Pusch, Kammerer & Fröbisch, 2019), the dorsal surface of the palate posterior to the internal choanae consists almost solely of the palatines, with components of the lacrimal laterally, the jugal posterolaterally and the pterygoid posteriorly and posteromedially (Figs. 5, 7).

As in the case of the vomer and palatines, the preserved portions of the pterygoids are laterally compressed, with the left pterygoid having been stretched as a result of anterior shifting of the left half of the skull, and the right pterygoid having been strongly compressed due to posterior shifting of the right half of the skull. The pterygoid forms the posterior portion of the palate, contacting the narrow posterior end of the vomer and the palatines anteriorly, and the posteroventral extension of the lacrimal and the anterior portion of the jugal anterolaterally. It bears a short transverse flange, which descends far down the medial surface of the dentary, with its ventral tip contacting the anterior portion of the prearticular. An ectopterygoid, which is usually situated at the lateral margin of the transverse flange of the pterygoid, could not be clearly delimited in the scan, but based on the condition in other basal cynodonts is likely to be present (Figs. 4, 7). Despite incompleteness of the posterior portions of the pterygoids, no interpterygoid vacuity seems to be present between them (Fig. 7), potentially indicating maturity in this specimen.

Lower jaw

The lower jaw of BSPG 1934-VIII-7 is tightly occluded to the upper jaw, and has suffered compression comparable to that of the cranium. Anteriorly, the left and right dentaries seem to be fused at the symphysis (Figs. 2, 3, 4, 8, 9). A fused mandibular symphysis, which is typically considered a eucynodont synapomorphy (Hopson & Kitching, 2001), has not yet been described for any cynodont stemward of that clade. Abdala et al. (2019) recently reported a fused mandibular symphysis for the Permian Vetusodon, but Huttenlocker & Sidor (2020) doubted the presence of this feature in this taxon and changed the coding of this character to ‘absent’ in their phylogenetic dataset. In BSPG 1934-VIII-7, a fused symphysis cannot be confirmed with certainty based on external examination, because of damage to the surface in the actual specimen. However, the CT-data (which shows no sign of a suture between the mandibular rami) and the fact that the jaw symphysis has remained in place as a single unit even after distortion (the dentaries frequently dislocate in therapsid skulls showing this style of compression) together provide strong evidence that this structure is fused in Bolotridon (Figs. 8, 9).

In lateral view, the dentaries gently curve upwards at the symphysis. They consist of a horizontal, tooth-bearing ramus, with its ventral margin being damaged on the right side, and a posterodorsally sloping coronoid process, which is slightly damaged on both sides. The tip of the coronoid process is located close to the postorbital bar and its posterior margin, which is better preserved on the right side, and slopes posteroventrally to contact the postdentary bones, of which only a portion of the right prearticular is preserved (Figs. 1–4, 8). The dentary houses three incisors (only the left incisor arcade is complete; the second incisor on the right side has fallen out), one canine, and ten postcanines. Both the canine and postcanines in the right dentary are preserved in place. In contrast, the left lower postcanine row is missing pc3 and pc4 and has been slightly shifted inwards due to lateral compression. The left pc1, pc2 and canine are placed somewhat anterior to their counterparts on the right side. As in the upper jaw, there is a short diastema present between the lower canine and first lower postcanine. The lower postcanines are generally poorly preserved, likely due to their appression and subsequent compression against the upper jaw. In the right dentary, both the last and penultimate postcanines are similar in size, but noticeably smaller than the other teeth with the exception of pc4. These two posteriormost teeth and pc4 either have damaged crowns or are not yet fully erupted (Fig. 8). The small holes visible on the ventral surface of the premaxilla just posterior to the upper incisor row are likely not caused by taphonomically-induced impression of the lower incisors into the upper jaw, but rather seem to be naturally present to accommodate the crowns of the enlarged lower teeth (Fig. 7B). Such holes for accommodating the lower incisors are also observed on the ventral surface of the premaxilla in Thrinaxodon (e.g. Jasinoski, Abdala & Fernandez, 2015: Fig. 7).

A well-developed masseteric fossa for the adductor musculature in the posterior portion of the horizontal ramus of the right dentary extends anteriorly to a point below the anterior margin of the orbit at the level between the last and penultimate upper postcanines, with its anterior and ventral margins being offset from the rest of the dentary by a pronounced lateral curvature around the postdentary bar (Fig. 2A). The definition and extension of the masseteric fossa, as well as the larger coronoid process with its tip being located close to the postorbital bar, closely resemble the condition described for Thrinaxodon and more derived cynodonts, rather than what is visible in the galesaurids Progalesaurus and Galesaurus in which the masseteric fossa is not well defined and the coronoid process is not as large, or Permian taxa in which the fossa is restricted to the coronoid process (Kemp, 1979; Abdala & Damiani, 2004; Sidor & Smith, 2004; Ivakhnenko, 2013; Kammerer, 2016; Huttenlocker & Sidor, 2020).

The splenial is present as a thin, anteroposteriorly elongated element positioned medial to the horizontal ramus of the right dentary by which it is completely obscured from lateral view. Anteriorly, it extends from the level of the root of the lower canine, where it contributes to the mandibular symphysis, to a level near the posterior margin of the dentary where it underlies the anterior tip of the prearticular (Figs. 4, 8A, 8C).

The right prearticular is broken and incomplete and only extends up to the posterior margin of the coronoid process. Its anterior portion, which contacts the splenial anteriorly, rests medial to the posterior margin of the dentary where it is overlapped by the ventral tip of the descending transverse flange of the pterygoid (Figs. 2, 4, 8).

The angular, which usually forms the ventral and posterolateral portion of the lower jaw, is also broken and incompletely preserved in BSPG 1934-VIII-7. Its anterior portion, which is only preserved on the left side, has been strongly shifted anteriorly extending medially along the tooth-bearing horizontal ramus of the dentary. An articular facet is visible on its posterodorsal surface to accommodate the prearticular, which is (with the exception of a small fragment) not preserved on the left side. The reflected lamina of the angular is completely missing in this specimen (Fig. 8).

Discussion

Phylogeny of early cynodonts

The parsimony analysis recovered 24 most parsimonious trees with a tree length of 256 (consistency index = 0.598, retention index = 0.809). Generally, the strict consensus and Bayesian inference majority rule topologies (Fig. 10) are similar to those of Huttenlocker & Sidor (2020), but the interrelationships of epicynodonts in our parsimony analysis are better resolved after including Bolotridon. Topology for non-epicynodont cynodonts is identical to that of Huttenlocker & Sidor (2020), with charassognathids and Dvinia recovered in a clade at the base of Cynodontia and Procynosuchus crownward of them.

Figure 10 Maximum parsimony strict consensus (left) and Bayesian inference majority rule (right) topologies of Permo-Triassic theriodonts obtained from the phylogenetic analyses.

Nodes of clades of interest are labeled: (A) Eutheriodontia; (B) Therocephalia; (C) Cynodontia; (D) Charassognathidae; (E) Abdalodontinae; (F) Epicynodontia; (G) Eucynodontia. Numbers at nodes represent clade support, if no bootstrap value is indicated on the left it was less than 50%, and Bayesian posterior probabilities on the right. Image by Luisa C. Pusch.

As in the earlier analysis, Abdalodon diastematicus and Nshimbodon muchingaensis are recovered as sister-taxa with extremely high support. These two taxa, the former known from a small, dorsoventrally crushed skull from the South African Endothiodon AZ and the latter from a relatively well-preserved skull from the Zambian Madumabisa Mudstone Formation, are exceedingly similar, and the diagnostic features of the latter cited by Huttenlocker & Sidor (2020) are all questionable. These authors differentiated Nshimbodon from Abdalodon by a gentler anteroventral slope of the rostrum and proportionally deeper dentary ramus. However, the diagnostic value of these characters is rendered dubious by extreme dorsoventral compression in the holotype of A. diastematicus, which has altered the snout profile and relative height of the dentary. Potential ontogenetic variation in jaw robusticity also should be considered, as the holotype of A. diastematicus represents a somewhat smaller specimen (5.61 cm skull length) than that of N. muchingaensis (7.0 cm). Other listed autapomorphies for Nshimbodon include tooth count and the presence of a series of neurovascular pits on the dentary, but in these features it accords closely with Abdalodon (Kammerer, 2016). Here, we synonymize Nshimbodon with Abdalodon, albeit tentatively retaining the species muchingaensis considering geographic and probable stratigraphic separation from A. diastematicus. Additional specimens from Zambia and South Africa will be required to determine whether even specific separation is warranted.

As in the analyses of Huttenlocker & Sidor (2020), Epicynodontia and Eucynodontia are recovered with strong support in our analyses. The strict consensus tree recovers the Triassic galesaurids Galesaurus and Progalesaurus at the base of Epicynodontia, followed by the Permian Cynosaurus (sometimes considered a galesaurid) and Vetusodon, here in a sister-taxon relationship and occupying a position crownward of the latter taxa. This contrasts with the strict consensus of Huttenlocker & Sidor (2020), in which the majority of non-eucynodont epicynodont taxa form an unresolved polytomy. In our Bayesian tree, galesaurids and Vetusodon + Cynosaurus form an unresolved polytomy at the base of Epicynodontia as in the Bayesian tree of Huttenlocker & Sidor (2020). The slightly more crownward position of Cynosaurus in our strict consensus tree differs from what was suggested by previous workers, who recovered Cynosaurus as stemward to Galesaurus and Progalesaurus (e.g., Abdala, 2007; Kammerer, 2016; Van den Brandt & Abdala, 2018; Abdala et al., 2019), but this position lacks unambiguous apomorphies supporting it. The clade formed by Vetusodon and Cynosaurus is recovered with low support in our parsimony analysis, but modest support in the Bayesian analysis.

Both the parsimony and Bayesian analyses recover Bolotridon as the sister-taxon of Eucynodontia (Fig. 10). Thrinaxodon is recovered in a polytomy with Nanictosaurus + Platycraniellus and Bolotridon + Eucynodontia in both analyses. The clade uniting Bolotridon + Eucynodontia has modest support in the parsimony analysis, but is recovered with strong support in the Bayesian analysis, despite many unknown characters as a result of the physical incompleteness of BSPG 1934-VIII-7. Evidence for a sister-taxon relationship between Bolotridon and Eucynodontia is provided by only a single but very important character: fusion of the mandibular symphysis, which was historically considered a eucynodont synapomorphy (Hopson & Kitching, 2001). A position outside of Eucynodontia is supported by the morphology of the dentary symphysis, which is low and gently sloping in Bolotridon, as in more stemward cynodonts. In eucynodonts the dentary symphysis is tall, steeply-sloped and forms a distinct ‘chin’. A similar morphology was also described for the dentary symphysis of the Permian Vetusodon and Cynosaurus, but this is here optimized as convergence (e.g., Van den Brandt & Abdala, 2018; Abdala et al., 2019; Huttenlocker & Sidor, 2020: Supplemental Information).

Comparison of Bolotridon with other cynodonts

The new data obtained from the CT-scan permit a better resolved position for Bolotridon than in the phylogenetic analysis of Sidor & Smith (2004), where it was nested in a polytomy with Galesauridae and a clade consisting of the more derived epicynodonts Thrinaxodon and Platycraniellus. Inclusion of Bolotridon in Galesauridae is not supported by our results. In addition to a fused mandibular symphysis, which is not known from any late Permian or Early Triassic cynodont, a crownward position for Bolotridon relative to the galesaurids is supported by three additional characters in our analysis. One of these characters uniting Bolotridon with the more derived epicynodonts Thrinaxodon, Nanictosaurus, Platycraniellus, and eucynodonts to the exclusion of galesaurids is the presence of a well-developed masseteric fossa extending to the angle of the horizontal ramus of the dentary. The other two characters show a more complex distribution, being shared with the crownward epicynodonts Thrinaxodon and Nanictosaurus but also stemward Permian cynodonts, and not with eucynodonts or galesaurids. These are the absence of a strongly recurved main cusp on the posterior postcanines (the presence of this character in Progalesaurus and Galesaurus is interpreted as having evolved convergently from eucynodonts (Huttenlocker & Sidor, 2020: Supplemental Information)) and the presence of a lingual cingulum in the lower postcanines. The absence of this feature in the lower postcanines of Progalesaurus and Galesaurus might have evolved independently from eucynodonts as well, but since this character is so far uncertain in Vetusodon, Cynosaurus, and Charassognathidae, this cannot be confirmed.

Although endocranial characters have been described for various therapsids in CT-assisted studies in recent years, with special focus on cynodonts as the therapsid clade ancestral to (and including) mammals (e.g., Hillenius, 1994; Kemp, 2009; Rodrigues, Ruf & Schultz, 2013, 2014; Rodrigues et al., 2018; Benoit, Manger & Rubidge, 2016; Benoit et al., 2017a, 2017b, 2017c, 2018, 2019; Araújo et al., 2017, 2018; Crompton, Musinsky & Owerkowicz, 2015; Crompton et al., 2017; Bendel et al., 2018; Pusch, Kammerer & Fröbisch, 2019; Pusch et al., 2020; Huttenlocker & Sidor, 2020), they still remain understudied compared to the well-known external craniodental features in synapsids and thus have been largely ignored in phylogenetic analyses. The data matrix of Huttenlocker & Sidor (2020) is the only matrix so far including newly adapted information from Benoit et al. (2017a) about the rarely preserved orbitosphenoid in therapsids. Despite damage and incomplete preservation of BSPG 1934-VIII-7, we were able to describe a few endocranial characters for Bolotridon that are consistent with an epicynodont position for this taxon. Those characters include a maxillary canal that would have arisen from a relatively small maxillary antrum (antrum visible in CT-scan, but canal itself not resolvable due to damage) and a possible maxilloturbinal ridge extending forward beyond the opening of the lacrimal. The presence of the latter is rendered somewhat uncertain by poor preservation and CT-resolution in this region of the snout; as mentioned in our description a ridge appears to be present but could also represent a crack. Ridges that would have served as attachment point for cartilaginous maxilloturbinals have been described so far only for the epicynodonts Galesaurus and Thrinaxodon and for eucynodonts such as the cynognathian Massetognathus and the probainognathian Probainognathus (Crompton, 2013; Crompton, Musinsky & Owerkowicz, 2015; Crompton et al., 2017; Pusch, Kammerer & Fröbisch, 2019); they are absent in the non-epicynodonts Abdalodon muchingaensis (Huttenlocker & Sidor, 2020) and Procynosuchus delaharpeae (Kemp, 1979). Based on its position crownward of Galesaurus and Thrinaxodon in our analysis, presence of a maxilloturbinal ridge can be inferred for Bolotridon, even if the current record is less than definitive. A relatively small maxillary antrum from which the branching maxillary canal extends forward has also been described for Galesaurus and Thrinaxodon and is additionally present in basal probainognathian eucynodonts such as Ecteninion and Lumkuia. This contrasts with what is observed in the cynognathians Trirachodon and Massetognathus, in which the antrum is massive by comparison, and more crownward probainognathians, in which the maxillary antrum is separated from the maxillary canal and resembles the mammaliaform pattern (Benoit, Manger & Rubidge, 2016; Benoit et al., 2019; Crompton et al., 2017; Pusch, Kammerer & Fröbisch, 2019).

However, data concerning the morphology of the maxillary canal and antrum of cynodonts stemward of Galesaurus are still lacking, so it is uncertain whether a relatively small antrum represents a derived condition among early cynodonts or if this feature is conservative among basal taxa of this clade. Denser sampling of endocranial features of basal cynodonts is needed to test where this and other mammalian features might have first evolved.

A unique fauna of small-bodied therapsids in the Trirachodon-Kannemeyeria Subzone

The Trirachodon-Kannemeyeria Subzone of the Cynognathus AZ is the most fossil- and species-rich of the three Cynognathus AZ subzones. Therapsids in this subzone are represented by the abundant dicynodont Kannemeyeria simocephalus, the rare therocephalians Microgomphodon oligocynus and Bauria cynops, and cynodonts such as the common Cynognathus crateronotus, Diademodon tetragonus, Trirachodon berryi, and Trirachodon/Cricodon kannemeyeri (the latter first appearing at the top of this subzone) (Abdala, Hancox & Neveling, 2005; Abdala & Ribeiro, 2010; Smith, Rubidge & Van der Walt, 2012: Table 2.5; Hancox, Neveling & Rubidge, 2020). In addition to the usual occurrences of Trirachodon, Cynognathus, Diademodon, and Kannemeyeria at Lady Frere, this area (consisting of both the Commonage and nearby localities like Lumku Mission in Chris Hani District Municipality) yields unique records of small-bodied therapsid taxa including the dicynodont Kombuisia frerensis, the basal cynodont Bolotridon frerensis, and the early probainognathian Lumkuia fuzzi (Broili & Schröder, 1934; Kitching, 1963; Hotton, 1974; Hopson & Kitching, 2001; Fröbisch, 2007; Abdala, Hancox & Neveling, 2005; Abdala & Ribeiro, 2010; Smith, Rubidge & Van der Walt, 2012: Table 2.5; Hancox, Neveling & Rubidge, 2020).

These seemingly anachronistic faunal elements (representing both ‘relict’ taxa like the emydopoid dicynodont Kombuisia and the non-eucynodont Bolotridon, and an early representative of the later-radiating cynodont clade Probainognathia in Lumkuia) highlight the complex and transitional nature of the Cynognathus AZ fauna. Gaps in our knowledge of small-bodied taxa throughout the Triassic make confident evaluation of the stratigraphic ranges of some groups difficult, complicating our understanding of what is truly ‘relictual’ vs. just rarely preserved. For example, the cynodonts Bolotridon and Lumkuia are not known from localities outside of the area surrounding Lady Frere, and are restricted to the Trirachodon-Kannemeyeria Subzone of the Cynognathus AZ. The type species of the rare dicynodont Kombuisia (K. frerensis), which is only known from two specimens from the Lady Frere locality, is also restricted to the Trirachodon-Kannemeyeria Subzone of the Cynognathus AZ. However, a second species of the genus, Kombuisia antarctica from the lower Fremouw Formation in Antarctica, extends the known stratigraphic range of Kombuisia into rocks that are equivalent in age to the Lower Triassic Lystrosaurus declivis AZ of South Africa (Fröbisch, 2007; Fröbisch, Angielczyk & Sidor, 2010; Botha & Smith, 2020; Hancox, Neveling & Rubidge, 2020).

It is possible that the apparent restriction of these taxa to the area surrounding Lady Frere is an artifact of unusual preservational conditions; something lending itself to the preservation of small-bodied, rare taxa not recorded at other Cynognathus AZ localities. Complicating this interpretation somewhat is the broader geographic range of certain otherwise-comparable coeval therapsids. The bauriid Bauria is also relatively small (basal skull length ~12 cm), is in some ways a ‘relict’ (being among the geologically youngest therocephalians), and occurs only in the Trirachodon-Kannemeyeria Subzone, but is much more widely distributed across the Karoo Basin (Kitching, 1963; Abdala et al., 2014; Hancox, Neveling & Rubidge, 2020). Another contemporaneous bauriid, Microgomphodon, exhibits substantially broader stratigraphic and geographic ranges, occurring from the lower Langbergia-Garjainia Subzone to the Trirachodon-Kannemeyeria Subzone in the South African Cynognathus AZ and also the upper Omingonde Formation of Namibia (Abdala et al., 2014; Hancox, Neveling & Rubidge, 2020). The greatest geographic and stratigraphic range for a Cynognathus AZ tetrapod, however, is that of the small parareptile Palacrodon browni. Originally known from the Cynognathus AZ of South Africa, this taxon was later discovered in the lower Fremouw Formation of Antarctica (which correlates chronostratigraphically with the Lower Triassic Lystrosaurus declivis AZ of South Africa) and the Upper Triassic Chinle Formation of northeastern Arizona, USA (Kligman, Marsh & Parker, 2018; Botha & Smith, 2020; Hancox, Neveling & Rubidge, 2020). The presence of Palacrodon in the middle Norian of the southwest United States extends the stratigraphic range of this genus by at least 15 million years, indicating remarkable persistence. Taken as a whole, these examples suggest that small-bodied Triassic tetrapods are greatly undersampled in the record. This is particularly likely to bias local estimates of diversity and richness, where localities may record the most abundant small-bodied taxa (e.g., trirachodontid cynodonts in the Cynognathus AZ), but with rarer small-bodied members of these assemblages unlikely to be preserved and/or collected. This is well-illustrated by the extreme paucity of known Palacrodon fossils, despite its stratigraphic duration of ~35 million years and Pangean distribution, and accentuated by the fact that the survival of emydopoid dicynodonts and non-eucynodont cynodonts into the Middle Triassic would be completely unknown if not for the exceptional record at Lady Frere. What makes this site exceptional is currently uncertain; whether the paleoenvironment favored by Kombuisia and Bolotridon was not present in other Cynognathus AZ localities, whether local depositional environments favored preservation of small-bodied animals, or whether some combination of these and/or collection bias explains its unusual local fauna is uncertain. Most of the published records from the area are from historical collections, and new, detailed lithological and stratigraphical studies are needed to understand the local environment and preservational regime in order to get answers.

Conclusions

Our redescription of the enigmatic basal cynodont Bolotridon frerensis, based on a computed tomographic (CT) reconstruction of the specimen BSPG 1934-VIII-7, provides new information on the palatal region and endocranial characters of this taxon that were previously obscured. Despite damage to the cranium, we were able to recognize several seemingly advanced craniodental characters shared with cynodonts more derived than galesaurids, including a well-developed masseteric fossa extending to the angle of the horizontal ramus of the dentary, a feature shared with more crownward epicynodonts such as Thrinaxodon and eucynodonts. Our recognition of a possibly fused mandibular symphysis in this taxon also supports a relatively crownward position for Bolotridon. The recovery of Bolotridon as the sister-taxon of Eucynodontia in both of our analyses indicates that while this taxon may be something of a relict, being from an earlier divergence than its contemporaries Cynognathus, Trirachodon, Diademodon and Lumkuia (Hancox, Neveling & Rubidge, 2020), it is not quite the Thrinaxodon-grade primordial cynodont that some previous authors have argued (e.g., Battail, 1991). It nevertheless represents part of an unusual assemblage for its time, and indicates that substantial diversity among Triassic taxa at small body size is likely currently unknown. Although this may in large part reflect real preservational bias, additional worker effort even in well-studied assemblages like the Cynognathus AZ is necessary, particularly at sites like Lady Frere known to produce rare, small-bodied taxa.

Supplemental Information

Supplemental Information 1 Characterlist used in this work.

Click here for additional data file.

Supplemental Information 2 Datamatrix used in this work.

Click here for additional data file.

Supplemental Information 3 Datamatrix used in this work.

Click here for additional data file.

We wish to thank Oliver Rauhut (Bayerische Staatssammlung für Paläontologie und Geologie, Munich, Germany) for the loan of BSPG 1934-VIII-7 and Kristin Mahlow (CT-Laboratory, Museum für Naturkunde, Berlin, Germany) for scanning the specimen. LCP would like to thank Mark MacDougall (Museum für Naturkunde, Berlin, Germany) and Neil Brocklehurst (University of Oxford, United Kingdom) for their brief introductions into PAUP* and MrBayes. Furthermore, we would like to thank Ken Angielczyk and Jim Hopson for their thorough and constructive reviews of the manuscript and Brandon Hedrick for work as editor.

Institutional abbreviations

BSPG Bayerische Staatssammlung für Paläontologie und Geologie, Munich, Germany.

NHMUK PV Natural History Museum, Vertebrate Palaeontological Collection, London, UK.

Additional Information and Declarations

Competing Interests

Author Contributions

Data Availability

The authors declare that they have no competing interests.

Luisa C. Pusch conceived and designed the experiments, performed the experiments, analyzed the data, prepared figures and/or tables, authored or reviewed drafts of the paper, and approved the final draft.

Christian F. Kammerer conceived and designed the experiments, performed the experiments, authored or reviewed drafts of the paper, contributed reagents/materials/analysis tools, and approved the final draft.

Jörg Fröbisch performed the experiments, authored or reviewed drafts of the paper, contributed reagents/materials/ analysis tools,, and approved the final draft.

The following information was supplied regarding data availability:

Raw CT scans are available at the Data Repository of the Museum für Naturkunde Berlin: DOI 10.7479/qcjn-dv44.

The data matrix and character list used in the phylogenetic analyses are available in the Supplemental Files.

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
