# Peer review of "Cranial anatomy of Bolotridon frerensis, an enigmatic cynodont from the Middle Triassic of South Africa, and its phylogenetic significance"

_PeerJ, doi:10.7717/peerj.11542_

## Round 0.1 · original submission · Minor Revisions

Dear authors,

Thank you for your submission to PeerJ. I agree with the reviewers that this manuscript is well-written and organized. The reviewers only had some minor changes to the text as well as possible figure changes. When you submit your revised manuscript, please include a clean version of the manuscript, a tracked changes version showing changes made, and an itemized reviewer response.

Let me know if you have any additional questions and I would be happy to answer them.

Best,

Brandon P. Hedrick, Ph.D.

·

Basic reporting

see below

Experimental design

see below

Validity of the findings

see below

Additional comments

General comments: This paper provides a new description of the basal cynodont Bolotridon. As the authors note, it has been somewhat overlooked compared to more common early cynodonts like Thrinaxodon. Using new CT scan data, they have been able to uncover new anatomical features that help resolve its phylogenetic position and potentially the evolution of some characters related to the presence of turbinals in the nasal cavity. The authors also note that Bolotridon was collected at a locality that has produced other atypical small taxa, and discuss the possible implications of this assemblage. The paper is well written and easy to read, with useful illustrations. It represents the kind of ct-based descriptive work that is needed to build up a database of comparative observations than can be incorporated into the next generation of phylogenetic analyses, and it definitely merits publication.

Overall, my suggestions for changes are minor, and most consist of suggestions for clarifications. In a couple of places, I noted cases where revisions to existing figures or additions of images would be helpful.

Line 27: I recommend changing strata to assemblage since you’re not specifically talking about a rock unit.

Line 39: change ‘currently instable’ to ‘previously unstable’

Line 53: I recommend adding a citation for Abdala and Ribeiro (2010) and Lukic-Walther e tal. (2019) as examples of more recent papers specifically about cynodont diversity patterns.

Lines 62-71: This is kind of a run-on sentence. I suggest dividing it up into two sentences, one that focuses on listing the other Karoo taxa, and the other focusing on the Karoo having the best record, potentially including evidence that the cynodont radiation began in the Permian.

Line 78: Maybe divide up the citations so that the ones pertaining to Galesaurids are earlier in the sentence than those pertaining to thrinaxodontids.

Line 89: I recommend replacing section with something like part or portion. I feel like using section here could cause confusion with an actual stratigraphic section at a locality, when that’s not the case.

Line 98: change while to whereas

Line 99: change its to the

Line 127: replace a first with novel

Line 164: Replace since with because. Also, later in this sentence, note which of the duplicate characters your deleted (presumably #64).

Line 170: A minor point, but is there a reason why you didn’t use a random addition sequence? It is unlikely to cause serious bias in the tree search with a fairly compact dataset like this one, but still a bit of a departure from common practice.

Line 183: Saying the septomaxilla is largely missing implies that some portion of it is still present. However, I don’t see it labeled in the figures, and there isn’t any subsequent description of it. Is there any preserved? If not, I recommend stating explicitly that it’s not preserved.

Line 192: change while to whereas

Line 231: I recommend specifying that the medial contact of the maxilla is on the ventral surface of the skull. It might seem obvious, but I think it’s good to be clear given that the approach each other fairly closely on the dorsal surface of the skull.

Line 244: I recommend moving the paragraph starting on line 263 (i.e., describing the internal and external surfaces of the maxilla) up so that it comes before the paragraph describing the dentition. To me it seems more logical to describe the all the features of the bone first, and then describe the dentition.

Line 246: Is there any explanation for the discrepancy in the Broili and Schroder’s tooth count (e.g., something that is easier to see with CT data)?

Line 260: change while to whereas

Line 283: It’s hard for me to see what you’re calling a ridge in Fig. 4. Anterior to the opening of the lacrimal canal, I see a somewhat rounded, raised structure. Is that what you’re referring to? If so, I think you should extend the leader line down so that it is in contact with it. If that’s not what you’re pointing out, I think you will need to add an image (either a cross section, or a new rendering, something like the one in Fig. 6) that shows the ridge more clearly. This is potentially a very important structure, so you want to make sure that readers can see its morphology clearly.

Line 302: The wording here makes it seem like you think the jugal/palatine contact is an artifact of deformation. If that is correct, I recommend saying that explicitly.

Line 315: I recommend labeling this in Fig. 6 and calling out the figure here.

Line 331: I realize that it probably doesn’t preserve much useful anatomy, but it seems odd that you don’t mention the parietal in the description of the skull roof. Even adding a sentence that says it doesn’t preserve much information would help avoid readers thinking it was omitted.

Line 336: remove only

Line 342: The absence of a grooved anterior section is somewhat hard to see in the figure. Is it possible to add a cross-section that shows this better?

Line 396: and ‘and’ before slopes

Line 398: change to: “...(only the left incisor arcade is complete; the second incisor on the right sidfe has fallen out)...”

Line 405: change mandible to dentary

Line 408: I think erupted might be a better word choice than developed here. I think this is particularly the case for PC 4. It seems to have a crown that is comparable in size to the others, it’s just not as far out of the dentary.

Line 421: I realize that you focus on galesaurids here given their closer temporal and likely phylogenetic proximity to Bolotridon, but it might also be good to add something like “and other basal cynodonts” to the sentence to help differentiate the condition here from what’s seen in the Permian taxa.

Line 479. I think you should start a new paragraph here. There seems to be a shift in main ideas from talking about characters uniting Bolotridon and eucynodonts above, to comparisons with more basal taxa from this line forward.

Line 534: subzones should not be capitalized.

Line 545: It seems like it would be appropriate to cite Hotton’s (1974) describing Kombusia here.

Ken Angeilczyk

·

Basic reporting

I have a few suggestions for changing the text listed below. The figures are well-done and the description in combination with the figures presents a good representation of the specimen.

line 27: "this strata" should be "these strata"
39: "instable" should be "unstable"
66: and [add "in"] Europe
67: .... suggesting that [add "the"] cynodont
87: . . . ., 2010) and perhaps Cistecynodon (Brink and Kitching, 19--).
95: taxon [add 'have] made it
103: "Crompton", not Compton
117: . . .505 [add . (period)] Most do not . . .
200: . . . level of [add "the fourth incisor"] (I4; Fig. 2, 3)
202: delete "but" and add "although" their apparent length . . .
223: median [delete"one" and substitute "ridge"]
231: . . . one another "in the palate" [delete "secondary"]
336: which appears [replace "only" with "to be"] very narrow
401 and elsewhere: The commonly-used convention of lower teeth being indicated by lower case letters (i, c, pc) and upper teeth by capitals (I, C, PC) is not observed in this paper, where all teeth are indicated by capitals. I do not suggest the authors redo the text and especially the figures for this paper, but suggest they cosider using the convention in subsequent papers.

Experimental design

The cladistic analysis is extremely well designed and I have no criticisms of it.

Validity of the findings

The authors are very cautious in their interpretation of the morphology of the specimen. They acknowledge repeatedly that interpreting this crushed specimen can yield differing interpretations of the restored morphology.

The cladogram of relationships among cynodonts is probably the most reliable one I know. The level of uncertainty with the interpretations are laid out, so we can estimate the probability of this particular configuration being the best possible given the uncertainties involved in interpreting less than complete morphology.

Additional comments

I would only note that the attribution of Bolotridon [Tribolodon of Seeley] to the family Galesauridae in Hopson and Kitching (1972) was in a far different (pre-cladistic) context of family-level relationships than what we accept now. We did not separate out a Thrinaxodontidae or Eucynodontidae, among other currently-accepted monophyletic groiups. So using our placement of early epicynodontians in the Galesauridae does not mean the same thing as it does now in a cladistic context. Perhaps this can be worked into the text.

---

## Round 0.2 · accepted · Accept

Dear Dr. Pusch,

Thank you for your careful attention to reviewer comments. I enjoyed reading this paper and thought that the description was exceptional in the way that it incorporated discussion of taphonomic distortion. I also thought that you integrated comparisons with other taxa quite effectively throughout. I am happy to accept this paper for publication in PeerJ.

I did notice a few grammatical/spelling issues in my final read-through that should be corrected prior to publication:

Line 202: I’m not sure what you mean by palatally? Towards the palate I’m guessing? Probably rewording this would be best.

Line 259: “The morphology of the intact left PC6 and right PC5 and PC6 appear to…’

Line 515: ‘with’ rather than ‘with with’

Line 531: ‘orbitosphenoid’ spelling

Thank you for your submission. Please contact me if you have any additional questions.

Best,

Brandon P. Hedrick, Ph.D.